# A Magnetic Photocatalytic Composite Derived from Waste Rice Noodle and Red Mud

**DOI:** 10.3390/nano15010051

**Published:** 2024-12-31

**Authors:** Qing Liu, Wanying Ying, Hailing Gou, Minghui Li, Ke Huang, Renyuan Xu, Guanzhi Ding, Pengyu Wang, Shuoping Chen

**Affiliations:** College of Materials Science and Engineering, Guilin University of Technology, Guilin 541004, China; 2120220336@glut.edu.cn (Q.L.); 2120210327@glut.edu.cn (W.Y.); 3212042041650@glut.edu.cn (H.G.); 2120230354@glut.edu.cn (M.L.); 3222042041416@glut.edu.cn (K.H.); 3212042041511@glut.edu.cn (R.X.); 1020220195@glut.edu.cn (G.D.); 2120220368@glut.edu.cn (P.W.)

**Keywords:** photocatalyst, waste materials, carbon quantum dots, maghemite

## Abstract

This study is the first to convert two waste materials, waste rice noodles (WRN) and red mud (RM), into a low-cost, high-value magnetic photocatalytic composite. WRN was processed via a hydrothermal method to produce a solution containing carbon quantum dots (CQDs). Simultaneously, RM was dissolved in acid to form a Fe^3+^ ion-rich solution, which was subsequently mixed with the CQDs solution and underwent hydrothermal treatment. During this process, the Fe^3+^ ions in RM were transformed into the maghemite (γ-Fe_2_O_3_) phase, while CQDs were incorporated onto the γ-Fe_2_O_3_ surface, resulting in the CQDs/γ-Fe_2_O_3_ magnetic photocatalytic composite. Experimental results demonstrated that the WRN-derived CQDs not only facilitated the formation of the magnetic γ-Fe_2_O_3_ phase but also promoted a synergistic interaction between CQDs and γ-Fe_2_O_3_, enhancing electron-hole pair separation and boosting the production of reactive radicals such as O_2_^·−^ and ·OH. Under optimized conditions (pH = 8, carbon loading: 10 wt%), the CQDs/γ-Fe_2_O_3_ composite exhibited good photocatalytic performance against methylene blue, achieving a 97.6% degradation rate within 480 min and a degradation rate constant of 5.99 × 10^−3^ min^−1^, significantly outperforming RM and commercial γ-Fe_2_O_3_ powder. Beyond methylene blue, this composite also effectively degraded common organic dyes, including malachite green, methyl violet, basic fuchsin, and rhodamine B, with particularly high efficiency against malachite green, reaching a degradation rate constant of 5.465 × 10^−2^ min^−1^. Additionally, due to its soft magnetic properties (saturation magnetization intensity: 16.7 emu/g, residual magnetization intensity: 2.2 emu/g), the material could be conveniently recovered and reused after photocatalytic cycles. Even after 10 cycles, it retained over 98% recovery and 96% photocatalytic degradation efficiency, underscoring its potential for cost-effective, large-scale photocatalytic water purification.

## 1. Introduction

Photocatalytic degradation is an efficient, safe, and eco-friendly water purification technology that uses semiconductor catalysts (e.g., titanium [1,2], zinc [3,4], tin [5,6], cadmium [7,8], or bismuth compounds [9,10,11]) and ultraviolet or solar light to break down organic pollutants in water into harmless byproducts like carbon dioxide, water, and nitrogen. Its advantages include high efficiency, low energy consumption, mild reaction conditions, and no secondary pollution [12,13,14]. However, most currently reported photocatalytic materials are non-magnetic powder catalysts, leading to a significant challenge after the photocatalytic water purification process: the difficulty in separating these materials from the treated water [15]. Furthermore, the high synthesis costs of catalysts pose additional challenges for large-scale industrial wastewater treatment [16].

Designing magnetic photocatalytic materials offers an effective solution to this challenge. These materials can be easily separated after the photocatalytic degradation process by applying an external magnetic field, enabling repeated use and supporting cost-effective, large-scale industrial water purification [17,18,19]. Among various magnetic particles, iron oxides, such as Fe_3_O_4_ [20], FeO [21], and Fe_2_O_3_ [22,23,24], have gained considerable attention in the design of magnetic photocatalytic materials due to their low-cost, abundant availability, and environmental friendliness. However, compared to classic photocatalysts like TiO_2_, ZnO, and SnO_2_, iron oxides generally exhibit lower photocatalytic activity, typically requiring appropriate doping or composite modifications to achieve practical photocatalytic degradation performance. Some studies have shown that incorporating carbon quantum dots (CQDs) can effectively enhance the photocatalytic activity of iron-based materials [25,26,27,28,29,30]. For instance, Li et al. synthesized CQDs-modified maghemite (γ-Fe_2_O_3_) photocatalysts via a solvothermal method, finding that CQDs significantly reduced the bandgap of γ-Fe_2_O_3_ and improved electron-hole separation rates, thereby enhancing the photocatalytic degradation of sulfamethoxazole [30]. Similarly, Hang et al. combined CQDs with goethite (α-FeOOH), achieving a CQDs/α-FeOOH photocatalytic composite with a tetracycline degradation efficiency 3.7 times higher than that of pristine α-FeOOH [25]. However, it is important to note that most reported CQD/ferrite photocatalytic composites often rely on commercial iron and carbon sources, resulting in relatively high synthesis costs that limit large-scale commercialization.

Cooking waste poses a significant ecological threat as it can infiltrate soil and aquatic ecosystems, presenting considerable challenges for urban management [31,32]. Current commercial cooking waste treatment technologies mainly include anaerobic digestion [33,34], aerobic composting [35], landfilling [36], incineration [37], and production of animal feed [38]. While these strategies can manage large volumes of cooking waste, they exhibit several evident drawbacks, including the need for substantial land, capital, and equipment investment, low product added value, and the potential generation of secondary pollutants like greenhouse gases and leachate [35,39]. To address these issues, our previous work proposed an innovative strategy to convert cooking waste like waste rice noodle (WRN) into high-value functional materials. This strategy first transforms WRN into carbon quantum dots (CQDs) via hydrothermal methods, subsequently combining these with specific inorganic nanoparticles such as TiO_2_ or ZnO to produce CQDs/TiO_2_ [40] or CQDs/ZnO [41] photocatalytic composites for water pollution control. These WRN-based CQDs/inorganic oxide composites demonstrated exceptional photocatalytic degradation efficiency under visible light irradiation, significantly outperforming commercial TiO_2_ or ZnO. However, due to their powdered nature, effective separation post-photocatalysis remains a challenge for practical industrial water treatment applications. Furthermore, we have successfully achieved the co-conversion of two types of waste, i.e., WRN and waste iron oxide scale (IOS), into cost-effective, magnetic carbon quantum dot/iron oxide (CQDs/FeO_x_) [42] photocatalytic composites for the first time. Compared to CQDs/TiO_2_ and CQDs/ZnO composites, the CQDs/FeO_x_ photocatalytic materials demonstrate significantly reduced costs while providing additional magnetic properties. This enables efficient separation after the photocatalytic cycle, facilitating effective reuse and supporting low-cost, large-scale photocatalytic water purification, thereby enhancing their commercial viability.

Compared to IOS, red mud (RM) is a more challenging metallurgical waste to handle. RM is a waste byproduct generated in the aluminum production industry during the extraction of alumina, typically resulting in 1.0 to 2.0 tons of RM for every ton of alumina produced [43]. As a major producer of alumina, China discharges over 100 million tons of RM annually, with cumulative stockpiles exceeding 1 billion tons [44,45]. Current handling methods rely primarily on the construction of RM storage facilities. However, these storage facilities not only consume significant land resources but also incur high maintenance costs. Furthermore, exposed RM can contribute to air pollution through weathering and dust, posing serious threats to public health and the ecological environment [46,47]. Although various strategies for resource recovery from RM exist, including using it to produce building materials, ceramics, fertilizers [48,49,50,51], and adsorbents, traditional RM-based products generally suffer from low added value, unstable quality, significant secondary pollution, and intense homogenization competition, failing to resolve the challenges of environmentally efficient resource utilization [52].

RM predominantly contains α-Fe_2_O_3_ and may contain trace amounts of TiO_2_, which has limited photocatalytic activity, along with a variety of non-photocatalytic impurities (such as garnet, sodalite, calcite, and aluminum oxides) that further reduce its overall photocatalytic efficacy. Moreover, RM lacks strong magnetic components like magnetite (Fe_3_O_4_), complicating recycling and reuse efforts. Therefore, RM is not readily applicable as a practical photocatalytic material. Current approaches to transforming RM into high-performance photocatalytic materials mainly involve two strategies. One is reacting RM with other materials to convert components like α-Fe_2_O_3_ into more active photocatalytic substances; for example, Shao et al. reported the synthesis of Bi_5_FeTi_3_O_15_ photocatalysts from RM in the presence of bismuth nitrate and titanium butoxide in a KCl-NaCl molten salt system [53]. The other is combining RM with other components to enhance its photocatalytic performance. For instance, Ren et al. demonstrated that composites of graphene oxide and RM could achieve a 79.8% degradation rate of tetracycline within 80 min, a notable improvement over RM alone [54]. However, most RM-based photocatalytic materials still require either expensive additional reagents or stringent synthesis conditions, and limited research has focused on their recyclability and reuse, hindering their commercial viability.

In our previous research, we proposed the concept that combining organic waste with inorganic waste to produce high-value functional materials can yield low-cost, environmentally friendly magnetic photocatalytic composites [42]. Reports suggest that trivalent iron ions (Fe^3+^) can be converted into γ-Fe_2_O_3_ under appropriate hydrothermal conditions, exhibiting both excellent magnetic properties and photocatalytic activity [55,56]. To address the aforementioned challenges in the design of RM-based photocatalytic materials, we explored a novel strategy that integrates the valorization of RM and WRN. This approach leverages the properties of CQDs derived from WRN as organic intermediates to transform RM into low-cost, magnetic, high-performance photocatalytic materials. Specifically, RM was first subjected to acid leaching to produce a Fe^3+^-containing solution, while WRN was hydrothermally treated to generate a CQD-containing solution. By mixing the CQD solution with the acid-leached RM solution and conducting hydrothermal reactions under controlled pH conditions, Fe^3+^ ions were converted into γ-Fe_2_O_3_ powder with magnetic and photocatalytic properties. Simultaneously, CQDs were anchored onto the surface of γ-Fe_2_O_3_, facilitating efficient electron-hole pair separation and suppressing recombination. This enhanced the generation of active radicals and significantly improved photocatalytic performance.

Through this strategy, we successfully achieved, for the first time, the synergistic transformation of two waste materials (RM and WRN) into cost-effective magnetic CQDs/γ-Fe_2_O_3_ photocatalytic composites under mild hydrothermal conditions without the use of expensive reagents. These composites not only exhibit excellent photocatalytic properties but are also easily recoverable via magnetic fields and capable of repeated use. This approach demonstrates significant potential for large-scale commercialization while providing a promising model for the co-valorization of diverse waste streams, aligning with the principles of sustainable development and green chemistry. This study investigates the composition, structure, photocatalytic degradation performance, magnetic properties, and recyclability of the resulting CQDs/γ-Fe_2_O_3_ photocatalytic composites, while also exploring the mechanisms underlying their photocatalytic degradation.

## 2. Experiment Section

### 2.1. Materials

In this study, the catering waste used, specifically waste rice noodle (WRN), was provided by the canteen of Guilin University of Technology (Guilin, China). Its primary organic component is starch (21.36 g/100 g), with minor amounts of protein (1.91 g/100 g) and fat (0.4 g/100 g). Red mud (RM, 200 mesh, primary elemental content: Fe 25.97 wt%; Ca 1.49 wt%; Ti 4.44 wt%; Al 16.26 wt%; Si 8.83 wt%; Na 9.99 wt%; O 33.03 wt%; C 5.01 wt%; K 0.31 wt%) was sourced from Shandong Aluminium Industry Co., Ltd, (Zibo, China). Other reagents used, including methylene blue (98.5% purity), malachite green (98% purity), methyl violet (98% purity), basic fuchsin (98% purity), rhodamine B (98% purity), hydrochloric acid (HCl, 99.9% purity), ammonia water (NH_3_·H_2_O, 99% purity), sodium hydroxide (NaOH), and maghemite (γ-Fe_2_O_3_, 98% purity), were all purchased from McClean Reagent (Shanghai, China) and were used without further purification.

### 2.2. Synthesis

The main process for the synergistic conversion of RM and WRN into the CQDs/γ-Fe_2_O_3_ composite is illustrated in Figure 1. First, WRN was used as the raw material, and CQDs solution was obtained by the hydrothermal treatment method previously reported by our group [40,41,42]. Detailed synthetic procedures can be found in Appendix A in the ESI. Meanwhile, 100 g of RM was mixed with 800 g of 10 mol/L hydrochloric acid, leached at 60 °C for 3 h, cooled, and filtered to obtain an RM leachate mainly containing Fe^3+^ ions. The primary metal ion composition of the RM leachate was determined using inductively coupled plasma optical emission spectrometry (ICP-OES), yielding the following concentrations: Fe 26.53 g·L^−1^, Ti 4.66 g·L^−1^, Al 9.67 g·L^−1^, Ca 2.56 g·L^−1^, and Na 3.57 g·L^−1^. The obtained RM leachate was then mixed with the CQDs solution at a specific volume ratio and stirred for 0.5 h. Subsequently, the pH of the mixture was adjusted to 4 or 8 (using ammonia) or 12 (using NaOH solution).

The mixture was then transferred to a Teflon-lined hydrothermal reactor (Kemi Instrument, Hefei, China), which was sealed and placed in an oven (Yuejin Medical Device, Shanghai, China) at 180 °C for a 6-h hydrothermal reaction. After cooling to room temperature, the mixture was centrifuged, and the resulting solid powder was washed repeatedly with distilled water and ethanol, a total of 6 times, then dried in an oven (Yuejin Medical Device, Shanghai, China) at 60 °C for 12 h. A brown powder, identified as the CQDs/γ-Fe_2_O_3_ composite, was obtained after grinding (See Figure 1). As shown in Table 1, the CQDs/γ-Fe_2_O_3_ composites were labeled from CF-1 to CF-7, based on the pH and the ratio of RM leachate to CQDs solution. The elemental content in these composites was analyzed, and the results are also presented in Table 1.

### 2.3. General Characterization and Measurement of Photocatalytic Performance

The CQDs/γ-Fe_2_O_3_ composite material underwent characterization and photocatalytic degradation testing in accordance with the procedures described in our earlier reports [40,41,42]. For comprehensive information, kindly consult Appendix A in the ESI.

## 3. Result and Discussion

### 3.1. Structural Characterization

In order to understand the interaction between the Fe^3+^ ions in RM leachate and CQDs under hydrothermal conditions, the raw RM and CQDs/γ-Fe_2_O_3_ composites were characterized by X-ray diffraction (XRD). As shown in Figure 2a, the iron phase in the raw RM was primarily composed of weakly magnetic α-Fe_2_O_3_, which is not suitable as a practical magnetic photocatalytic material. However, after reacting the RM leachate containing Fe^3+^ ions with CQDs, the main crystalline phase of the resulting material transformed into magnetic γ-Fe_2_O_3_. The diffraction peaks at 18.4°, 30.2°, 35.6°, 43.3°, 53.7°, 57.3°, and 62.9° corresponded to the (111), (220), (311), (400), (422), (511), and (440) crystal planes of γ-Fe_2_O_3_ (maghemite, PDF card No. 00-039-1346). Compared to the primary phase α-Fe_2_O_3_ in the raw RM material, γ-Fe_2_O_3_ exhibits stronger magnetic properties and demonstrates significant photocatalytic activity when combined with CQDs. This is of great importance for the development of practical magnetic photocatalytic materials.

Since the initial RM leachate is strongly acidic, it is necessary to adjust the pH value to obtain the CQDs/γ-Fe_2_O_3_ composite. The appropriate pH value is one of the key factors for achieving a higher purity γ-Fe_2_O_3_ phase. As shown in Figure 2b and Table 1, when the reaction system was adjusted to weakly acidic conditions (pH = 4, CF-1 sample) using ammonia, the γ-Fe_2_O_3_ magnetic phase was obtained. However, the γ-Fe_2_O_3_ content was relatively low, containing only 51.9 wt% γ-Fe_2_O_3_. The diffraction patterns also revealed the presence of a large amount of catalytically inactive aluminum oxyhydroxide (AlO(OH), PDF card No. 00-039-1346) in the CF-1 sample. By adjusting the pH to near-neutral conditions (pH = 8) using ammonia (CF-2 sample), a relatively pure γ-Fe_2_O_3_ phase with content as high as 80.7 wt% was obtained, while the diffraction peaks of impurity phases such as AlO(OH) became insignificant. Adjusting the pH to strongly alkaline conditions (pH = 12, CF-3 sample) using sodium hydroxide also resulted in a relatively pure γ-Fe_2_O_3_ phase (79.7 wt%). However, the diffraction peak intensity was significantly weaker compared to the CF-2 sample obtained under near-neutral conditions, possibly due to the poorer crystallinity of the γ-Fe_2_O_3_ phase formed in the strongly alkaline environment.

On the other hand, the presence of CQDs was crucial for ensuring the formation of the γ-Fe_2_O_3_ phase. As shown in Figure 2c and Table 1, in the absence of CQDs, the RM leachate in a pH = 8 environment could only be reconverted to α-Fe_2_O_3_ instead of γ-Fe_2_O_3_, while a significant amount of the impurity phase AlO(OH) was also generated (CF-4 sample). It was only through the introduction of CQDs that the γ-Fe_2_O_3_ magnetic phase could be formed, effectively suppressing the incorporation of impurity phases such as AlO(OH) into the photocatalytic material (CF-2, CF-5 to CF-7 samples). However, when an excessive amount of CQDs was added (CF-6 and CF-7 samples), the diffraction peaks of the γ-Fe_2_O_3_ phase were significantly weakened.

We also found that both pH and the dosage of CQDs could influence the particle size distribution of the final product. As shown in Figure 3a, with the increase in hydrothermal reaction pH, the median particle size (D50) of the CQDs/γ-Fe_2_O_3_ composite first decreased and then increased. The CF-2 sample obtained in the pH = 8 environment exhibited the smallest D50 value (6.399 μm). Similarly, Figure 3b shows that with the introduction of CQDs, the D50 of the CQDs/γ-Fe_2_O_3_ composite also followed a trend of first decreasing and then increasing as the CQD loading increased. The CF-2 sample with a CQDs loading of approximately 10 wt% exhibited the smallest D50 value. Smaller particle sizes may result in a larger specific surface area, facilitating the exposure of active crystal facets, which is beneficial for enhancing the photocatalytic degradation activity of the product.

As shown in Figure 4a, the overall microstructure of the CQDs/γ-Fe_2_O_3_ composite material presented as irregular sheets. The EDS analysis revealed a uniform distribution of C, Fe, and O elements on the surface of the composite (See Figure 4b). TEM analysis of the CQDs/γ-Fe_2_O_3_ composite (CF-2 sample) in Figure 4c showed that spherical CQDs particles were evenly dispersed on the surface of γ-Fe_2_O_3_. The HRTEM image in Figure 4d displayed the interlaced lattice of γ-Fe_2_O_3_ and CQDs, where the lattice fringe spacing of 0.250 nm corresponded to the (311) crystal plane of γ-Fe_2_O_3_, while the spacing of approximately 0.283 nm was attributed to the (020) crystal plane of CQDs. In summary, it could be confirmed that the CQDs derived from WRN were successfully combined with γ-Fe_2_O_3_ derived from RM in the resulting composite.

The infrared spectrum showed that the surface of the CQDs/γ-Fe_2_O_3_ composite contained a large number of hydroxyl groups originating from CQDs. The stretching and bending vibrations of the O–H bond appeared at 3310 cm^−1^ and 1630 cm^−1^, respectively, while the stretching vibration peak of C–O was observed at 1060 cm^−1^. The commercial γ-Fe_2_O_3_ exhibited three distinct Fe–O stretching vibration peaks in the low wavenumber region below 700 cm^−1^. Among these, the peaks at 575 cm^−1^ and 636 cm^−1^ were attributed to the υ_1_ band of Fe–O, while the peak at 440 cm^−1^ was assigned to the υ_2_ band of Fe–O [57,58]. In comparison, after the incorporation of CQDs, the Fe–O stretching vibration peaks in the CQDs/γ-Fe_2_O_3_ composite underwent significant shifts. Specifically, the peaks at 440 cm^−1^ and 575 cm^−1^ blue-shifted to 478 cm^−1^ and 590 cm^−1^, respectively, while the peak at 636 cm^−1^ red-shifted to 617 cm^−1^. These observations suggested that during the hydrothermal reaction, the surface hydroxyl groups of the generated γ-Fe_2_O_3_ reacted with the hydroxyl groups of the CQDs, forming Fe–O–C bonds. This reaction resulted in the tight binding between CQDs and γ-Fe_2_O_3_, leading to the formation of a stable CQDs/γ-Fe_2_O_3_ composite (Figure 5a,b). The Raman spectrum (see Appendix A in ESI) further supported the successful synthesis of the CQDs/γ-Fe_2_O_3_ composite. The Raman scattering peaks at 1350 cm^−1^ and 1569 cm^−1^ correspond to the D-band and G-band of the carbon structure in the CQDs, respectively. The Raman peak at 1000 cm^−1^ could be attributed to the vibration of C–O bonds on the surface of the CQDs, while the peak at 382 cm^−1^ could be assigned to the stretching vibration of Fe–O bonds. Additionally, although the surface of the CQDs/γ-Fe_2_O_3_ composite was rich in hydroxyl groups, the electron transfer effect induced by the coupling of CQDs with γ-Fe_2_O_3_ caused the hydroxyl groups to favor their non-dissociated state, thereby reducing the acidity of the surface hydroxyl groups in the composite system. As a result, the PZC value of the CQDs/γ-Fe_2_O_3_ composite (8.54) was higher than that of the non-composite γ-Fe_2_O_3_ powder (8.09) (Figure 5c).

The tight binding between CQDs and γ-Fe_2_O_3_ in the composite material was further confirmed by XPS analysis (Figure 5d–g). Figure 5d shows the presence of Fe, O, and C elements in the CQDs/γ-Fe_2_O_3_ composite. As shown in Figure 5e and Table 2, the Fe 2p spectrum of commercial γ-Fe_2_O_3_ exhibited characteristic peaks at 723.68 eV, 717.81 eV, 712.66 eV, and 710.05 eV, respectively. The peak at 723.68 eV was assigned to the Fe(2p_1/2_) signal, while the peaks at 712.66 eV and 710.05 eV were attributed to the Fe(2p_3/2_) signal [59,60,61]. Notably, the Fe(2p_3/2_) signal showed a separation between the Fe^2+^ and Fe^3+^ signals, indicating the presence of both Fe^2+^ and Fe^3+^ ions, with the Fe^2+^ ions likely originating from the surface or crystal defect regions of γ-Fe_2_O_3_ [62,63]. The peak at 717.81 eV was assigned to the Fe^3+^ satellite peak. After the incorporation of CQDs, the Fe 2p spectrum of the CQDs/γ-Fe_2_O_3_ composite displayed a shift of the Fe^3+^ satellite peak to a higher binding energy region (719.86 eV), and a similar phenomenon was observed for the Fe(2p_1/2_) signal peak. This suggested C–O–Fe interaction between the Fe^3+^ on the surface of γ-Fe_2_O_3_ and the CQDs leads to charge transfer and redistribution of the electron density of Fe [28,64]. Meanwhile, the high-resolution O 1s spectrum of γ-Fe_2_O_3_ powder showed two peaks at 530.66 eV and 529.21 eV, which were attributed to Fe–O bonds and surface hydroxyl groups of iron oxides, respectively. In contrast, after the incorporation of CQDs, the O 1s spectrum of the CQDs/γ-Fe_2_O_3_ composite displayed characteristic peaks at 530.19 eV and 531.79 eV, corresponding to Fe–O and C–O bonds, respectively (Figure 5f and Table 2). Combined with the IR results, it was inferred that the composite reaction between CQDs and γ-Fe_2_O_3_ involved the reaction between hydroxyl groups on CQDs and surface hydroxyl groups on γ-Fe_2_O_3_, leading to the disappearance of the iron oxide hydroxyl signal and the appearance of the C–O bond signal. Additionally, the high-resolution C 1s spectrum of the CQDs/γ-Fe_2_O_3_ composite showed a characteristic peak at 284.86 eV, which was attributed to graphitic carbon in the CQDs, while the peaks centered at 286.08 eV, 287.10 eV, and 288.67 eV corresponded to the C–O bond, C–O–Fe bond, and aromatic ring of CQDs, respectively (Figure 5g and Table 2). Thus, it was concluded that γ-Fe_2_O_3_ was successfully combined with CQDs.

### 3.2. Magnetic Properties of CQDs/γ-Fe_2_O_3_ Composites

The magnetically responsive photocatalytic powder material was capable of being swiftly recovered through magnetic fields upon completion of the photocatalytic degradation process, allowing for repeated usage, thereby reducing costs and preventing residual contamination in environmental water bodies that could lead to secondary pollution. This imparted it a distinct competitive advantage in practical industrial water purification applications. As illustrated by the magnetization curves in Figure 6a, due to the introduction of CQDs enabling the formation of the magnetic γ-Fe_2_O_3_ phase, the CQDs/γ-Fe_2_O_3_ composite material exhibited excellent soft magnetic properties. Its saturation magnetization reached 16.7 emu/g, while also exhibiting a relatively low remanent magnetization (2.2 emu/g) and coercivity (38.9 Oe). It was noteworthy that without the introduction of CQDs, the resulting product (CF-4 sample) mainly consisted of α-Fe_2_O_3_, which exhibited almost no magnetic properties and could not be efficiently recovered using a magnetic field.

The magnetism of the resulting CQDs/γ-Fe_2_O_3_ composite ensured that the powder dispersed in water or other media could rapidly aggregate under a magnetic field, enabling convenient recovery. As shown in Figure 6b,c, after completing the photocatalytic degradation of methylene blue, the resulting CQDs/γ-Fe_2_O_3_ composite could be recovered via magnetic fields and reused, with a recovery rate as high as 99.8% after the first cycle and still maintaining a recovery rate of 98.0% even after 10 photocatalytic cycles.

### 3.3. Photocatalytic Performance of CQDs/γ-Fe_2_O_3_ Composites

In addition to its favorable magnetic properties, the synthesized CQDs/γ-Fe_2_O_3_ composite material demonstrated excellent photocatalytic degradation performance for pollutants such as organic dyes. As shown in Figure 7a–c, methylene blue, a dye with significant toxicity to aquatic organisms, exhibited strong resistance to degradation under 405 nm UV light. Neither RM nor commercial γ-Fe_2_O_3_ used as catalysts showed notable photocatalytic activity toward methylene blue. For instance, in the presence of RM, the degradation rate of methylene blue reached only 26.5% after 8 h (480 min) of photocatalysis, while commercial γ-Fe_2_O_3_ achieved a degradation rate as low as 14.8%. In contrast, the CQDs/γ-Fe_2_O_3_ composite (CF-2 sample) effectively mitigated methylene blue pollution, achieving a photocatalytic degradation rate of 97.6% within 480 min, with a degradation rate constant of 5.99 × 10^−3^ min^−1^, indicating near-complete degradation of methylene blue. It was noteworthy that the CQDs/γ-Fe_2_O_3_ composite exhibited a good adsorption effect on methylene blue. However, adsorption alone could not completely remove methylene blue (with a removal rate of only 48.1% within 8 h), indicating that photocatalysis remained the primary method for achieving high-efficiency removal of methylene blue. Nonetheless, the high adsorption activity of the CQDs/γ-Fe_2_O_3_ composite also effectively promoted its photocatalytic performance. Moreover, the material’s magnetic properties facilitated convenient recovery via magnetic separation, enabling efficient recycling and reuse. Even after ten photocatalytic cycles, the degradation efficiency of methylene blue remained at 96.6%, as shown in Figure 7d.

During synthesis, two critical factors significantly influenced the photocatalytic performance of the CQDs/γ-Fe_2_O_3_ composite. One factor was the pH during the hydrothermal reaction. As depicted in Figure 8a–c, the CF-1 sample, prepared under weakly acidic conditions (pH = 4), contained substantial AlO(OH) impurities. Although this sample achieved the formation of γ-Fe_2_O_3_ and CQDs composites, its photocatalytic activity was significantly lower than that of samples synthesized under near-neutral (pH = 8, CF-2) or strongly alkaline conditions (pH = 12, CF-3). The CF-1 sample degraded only 75.1% of methylene blue after 480 min of illumination, with a degradation rate constant of 1.54 × 10^−3^ min^−1^. Higher pH levels promoted the formation of the γ-Fe_2_O_3_ phase and suppressed the formation of impurity phases such as AlO(OH), thereby enhancing the photocatalytic performance. The CF-2 sample, synthesized at pH = 8, exhibited a γ-Fe_2_O_3_ phase with both higher crystallinity and purity, as well as smaller particle size, and thus exhibited optimal photocatalytic activity, achieving complete degradation of MB within 480 min with a rate constant of 5.99 × 10^−3^ min^−1^. In contrast, the CF-3 sample, prepared under strongly alkaline conditions, achieved a 92.3% degradation rate for MB within the same period. This slight reduction in performance was attributed to the weakened crystallinity and larger particle size of γ-Fe_2_O_3_ under highly alkaline conditions, which reduced the exposure of active crystal facets [65].

On the other hand, the introduction of WRN-derived CQDs not only facilitated the formation of the magnetic γ-Fe_2_O_3_ phase, enabling convenient recovery, but also enhanced the photocatalytic performance of the CQDs/γ-Fe_2_O_3_ composite through their strong coupling with the γ-Fe_2_O_3_ phase. Figure 8d–f show that without the incorporation of CQDs, RM leachate predominantly transformed into a mixture of non-magnetic α-Fe_2_O_3_ and AlO(OH), which exhibited negligible photocatalytic degradation effects on methylene blue under 405 nm UV light. Only the introduction of WRN-derived CQDs endowed the composite with effective photocatalytic degradation capabilities. The photocatalytic efficiency of the CQDs/γ-Fe_2_O_3_ composite varied with the CQDs loading amount, initially increasing and then decreasing. The CF-4 sample, containing approximately 10 wt% CQDs, demonstrated the best photocatalytic performance. In contrast, excessive CQD loading suppressed the photocatalytic activity. For instance, the CF-7 sample, with around 25 wt% CQDs, exhibited a degradation rate constant of only 2.62 × 10^−3^ min^−1^, less than half that of the CF-2 sample. This suppression was attributed to the shielding effect [66] of the carbon component and the reduced crystallinity, as well as the excessively large particle size of γ-Fe_2_O_3_ caused by the excess CQDs [65].

As shown in Figure 9, the synthesized CQDs/γ-Fe_2_O_3_ photocatalytic composite (CF-2 sample) exhibited effective photocatalytic degradation capabilities not only for methylene blue but also for water-soluble dyes based on the triphenylmethane structure, such as malachite green, methyl violet, basic fuchsin, and rhodamine B. Its photocatalytic performance was superior to that of commercial γ-Fe_2_O_3_ powder. It was also observed that the molecular structure of the dyes influenced their degradability. Among triphenylmethane dyes, those with dimethylamino substituents, such as malachite green, were most readily degraded, achieving a degradation rate of 96.4% within 1 h and a high degradation rate constant of 5.465 × 10^−2^ min^−1^. In contrast, dyes containing amino groups as substituents, such as methyl violet and basic fuchsin, exhibited significantly lower photocatalytic degradation efficiency. However, nearly complete degradation was still achieved within approximately 4 h. For triphenylmethane dyes containing carboxyl groups, such as rhodamine B, degradation was relatively more challenging. For a 20 mg/L rhodamine B solution, the CQDs/γ-Fe_2_O_3_ composite achieved a degradation rate of 88.3% after approximately 5 h.

As shown in Table 3, the CQDs/γ-Fe_2_O_3_ composite synthesized in this study exhibited superior photocatalytic performance compared to other photocatalytic composites utilizing different commercial iron or carbon sources [25,26,27,28,29,30,67,68]. Additionally, the exclusive use of waste materials (WRN and RM) as raw material sources significantly reduced the synthesis cost compared to other iron-based photocatalytic composites, making the CQDs/γ-Fe_2_O_3_ composite more suitable for low-cost, large-scale water purification projects. This approach also delivers considerable environmental benefits, aligning with the principles of sustainable development. Compared to previously reported CQDs/FeO_x_ composites derived from WRN and waste IOS [42], the CQDs/γ-Fe_2_O_3_ composite in this study achieved comparable photocatalytic degradation activity. However, unlike IOS, RM is a metallurgical waste with a more complex composition, greater pollution risk, and more challenging disposal requirements. The synergistic conversion of RM and organic waste WRN into a CQDs/γ-Fe_2_O_3_ composite with practical photocatalytic degradation capability and magnetic recoverability addresses a critical challenge in the high-value utilization of RM. This approach offers a novel solution with greater environmental and sustainable development significance.

Compared to the previously reported CQDs/TiO_2_ [40] or CQDs/ZnO [41] composites based on WRN, the CQDs/γ-Fe_2_O_3_ composite based on WRN and RM exhibited relatively poor photocatalytic activity. This was somewhat related to its inorganic phase being iron oxide. However, the CQDs/γ-Fe_2_O_3_ composite possessed the advantage of being easily recoverable via a magnetic field, which enabled reliable recycling and prevented secondary pollution. Moreover, since its raw materials were entirely derived from waste, it could be produced at a very low cost while effectively achieving high-value recycling of waste, aligning with the principles of a green circular economy and offering stronger market competitiveness (See Table 4). Additionally, the design system for such materials still has significant potential for expansion. It can be anticipated that by coating or doping small amounts of Ti or Zn compounds onto the CQDs/γ-Fe_2_O_3_ composite, the photocatalytic degradation activity could be further enhanced while maintaining its magnetic photocatalytic properties.

### 3.4. Photocatalytic Mechanism of CQDs/γ-Fe_2_O_3_ Composites

To further elucidate the mechanism by which WRN-derived CQDs enhance the photocatalytic performance of the CQDs/γ-Fe_2_O_3_ composite, a series of tests were conducted on the CF-2 sample, which demonstrated optimal photocatalytic activity with commercial γ-Fe_2_O_3_ powder used as a reference. As shown in Figure 10a, UV diffuse reflectance spectra revealed that compared to pure γ-Fe_2_O_3_, the introduction of CQDs enabled the CQDs/γ-Fe_2_O_3_ composite to exhibit stronger absorption across the entire visible light spectrum, improving its ability to utilize visible light energy. Bandgap width analysis (Figure 10b) showed that the CQDs/γ-Fe_2_O_3_ composite exhibited a significantly narrowed bandgap (approximately 1.27 eV) compared to commercial γ-Fe_2_O_3_ powder (1.57 eV), indicating that CQD incorporation reduced the bandgap, thereby enhancing the utilization of visible light energy, promoting electronic transitions, and facilitating charge separation. Fluorescence spectroscopy (Figure 10c) revealed that the fluorescence emission intensity of the CQDs/γ-Fe_2_O_3_ composite was substantially lower than that of γ-Fe_2_O_3_, indicating that CQDs not only facilitated charge separation but also effectively reduced the recombination of photogenerated electrons and holes, thereby enhancing photocatalytic performance. This was further corroborated by photocurrent and impedance analyses. As shown in Figure 10d,e, under UV light irradiation, the photocurrent intensity of the CQDs/γ-Fe_2_O_3_ composite was approximately 21 times higher than that of commercial γ-Fe_2_O_3_ powder, demonstrating more efficient interfacial charge transfer and reduced electron–hole recombination. Additionally, the Nyquist plot showed that the CQDs/γ-Fe_2_O_3_ composite had a smaller semicircle diameter than pure γ-Fe_2_O_3_, reflecting a lower charge transfer resistance, which ensured more effective charge separation and higher photocatalytic degradation activity. Furthermore, as shown in Figure 10f, the incorporation of CQDs significantly increased the specific surface area of the CQDs/γ-Fe_2_O_3_ composite, enhancing the adsorption of organic pollutants and contributing to improved photocatalytic performance.

XPS valence band (VB) spectroscopy was employed to analyze the VB potentials of the CQDs/γ-Fe_2_O_3_ composite and pure γ-Fe_2_O_3_. As shown in Figure 11a, the VB potential of pure γ-Fe_2_O_3_ was approximately 2.5 eV, higher than the standard electrode potential E^0^(·OH, H^+^/H_2_O) for hydroxyl radicals (2.38 eV vs. NHE). Combined with its bandgap width (1.57 eV), the conduction band (CB) potential of pure γ-Fe_2_O_3_ was calculated to be 0.93 eV, higher than E^0^(O_2_/O_2_^·−^) for superoxide radicals (−0.33 eV vs. NHE). This indicated that while pure γ-Fe_2_O_3_ could generate hydroxyl radicals (·OH) through photogenerated holes, it could not reduce oxygen to form superoxide radicals (O_2_^·−^). In contrast, after the introduction of CQDs, the resulting CQDs/γ-Fe_2_O_3_ composite could be considered a direct Z-Scheme heterojunction [41,69], which exhibited a higher VB potential of 2.75 eV, thereby facilitating the oxidation of water to produce ·OH radicals. Combined with its own bandgap width of 1.40 eV and the intrinsic bandgap of CQDs (2.08 eV), its CB potential was calculated to be −0.56 eV, which was lower than E^0^(O_2_/O_2_^·−^) (−0.33 eV vs. NHE), indicating its ability to reduce oxygen and generate O_2_^·−^ radicals. 

The roles of different reactive species in methylene blue degradation were further investigated using scavengers (EDTA-2Na, IPA, and BQ). Results showed that adding EDTA-2Na slightly reduced photocatalytic efficiency, retaining 74.3% of the original activity, suggesting that photogenerated holes played a minor role. In contrast, IPA and BQ significantly suppressed degradation, retaining only 40.5% and 40.9% of the activity, respectively, indicating that ·OH and O_2_^·−^ radicals were the primary reactive species (Figure 11b). Electron spin resonance (ESR) spectroscopy, using DMPO as a radical scavenger, further confirmed this. As shown in Figure 11c,d, characteristic peaks of ·OH and O_2_^·−^radicals were observed in water and methanol solutions, respectively, for the CQDs/γ-Fe_2_O_3_ composite, whereas pure γ-Fe_2_O_3_ generated negligible signals under identical conditions (Figure 11e,f). In summary, the introduction of WRN-derived CQDs enhanced the photocatalytic performance of the CQDs/γ-Fe_2_O_3_ composite through multiple mechanisms. Coupled with the γ-Fe_2_O_3_ substrate, the system effectively utilized visible light energy and adsorbed pollutant molecules. Upon visible light irradiation, the composite facilitated the excitation of electrons to the CB while leaving holes in the VB. The excited electrons were rapidly transferred between CQDs and γ-Fe_2_O_3_, suppressing recombination and promoting charge separation. Photogenerated holes reacted with water to produce ·OH radicals, while photogenerated electrons reacted with oxygen to produce O_2_^·–^ radicals, both of which effectively degraded various organic pollutants, demonstrating excellent photocatalytic activity (Figure 11g).

Using methylene blue as the target pollutant, we conducted a preliminary investigation into the possible degradation pathways of methylene blue via CQDs/γ-Fe_2_O_3_ composite. LC-MS was employed to analyze the intermediate products formed during the degradation process. Figure 12a,b display the HPLC chromatograms of methylene blue solution after 1 h and 8 h of photocatalytic degradation. For the solution degraded for 1 h, the most intense peak appeared at a retention time of 9.9 min (*m*/*z* = 105), corresponding to the degraded methylene blue molecule. Additionally, a peak at 6.5 min (*m*/*z* = 318) emerged, which can be attributed to a hydroxyl radical-addition product of the methylene blue aromatic ring [70]. Peaks corresponding to smaller molecular weight compounds were observed at 9.6 min (*m*/*z* = 171) and 1.0 min (*m*/*z* = 233), representing phenolic compounds and sulfonic acids, respectively, which are likely products of the hydroxylated methylene blue (*m*/*z* = 318) resulting from cleavage of the phenothiazine structure. After 8 h of photocatalytic degradation, the chromatographic peak at 9.9 min for methylene blue almost disappeared, along with the peak at 1.0 min. The peaks at 9.6 min and 6.5 min showed significant attenuation, and three new small-molecule peaks emerged at retention times of 1.1 min (*m*/*z* = 104), 5.4 min (*m*/*z* = 114), and 9.1 min (*m*/*z* = 125). These newly observed peaks are likely secondary products formed from further decomposition of phenolic intermediates. Based on these results, a possible degradation pathway for methylene blue under CQDs/γ-Fe_2_O_3_ composite photocatalysis is proposed, as illustrated in Figure 12c. Initially, hydroxyl radicals attack the aromatic rings of methylene blue, forming unstable intermediates. Subsequently, the phenothiazine core of methylene blue undergoes cleavage under the action of hydroxyl and superoxide radicals, yielding sulfonic acids or phenolic small molecules. These intermediates can further degrade into smaller phenols, polyols, or ketone compounds, which eventually oxidize into non-toxic molecules or ions, such as carbon dioxide, water, nitrate, and sulfate.

## 4. Conclusions

This study represents the first successful demonstration of the simultaneous conversion of an organic waste material (WRN) and a metallurgical waste material (RM) into CQDs/γ-Fe_2_O_3_ composite with excellent photocatalytic degradation activity and magnetic properties. The key to this preparation strategy was the hydrothermal reaction conducted by introducing WRN-derived CQDs into Fe^3+^-containing RM leachate under optimized pH conditions. This process facilitated the transformation of Fe^3+^ ions into the magnetic γ-Fe_2_O_3_ phase while enabling the CQDs to bind to the surface of γ-Fe_2_O_3_, forming heterojunctions. The coupling of CQDs with γ-Fe_2_O_3_ effectively promoted electron–hole pair separation and suppressed their recombination, leading to the generation of abundant ·OH and O_2_^·−^ radicals, which contributed to outstanding photocatalytic degradation of multiple dyes. Additionally, the CQDs/γ-Fe_2_O_3_ composite retained the magnetic properties imparted by γ-Fe_2_O_3_, allowing for rapid and efficient magnetic separation after photocatalytic cycles. This feature ensures cost-effective reuse and supports large-scale, low-cost applications in photocatalytic water purification. The CQDs/γ-Fe_2_O_3_ composite addresses the industry’s need for affordable, recyclable photocatalysts, showcasing significant commercialization potential. However, due to the inorganic phase being γ-Fe_2_O_3_, the CQDs/γ-Fe_2_O_3_ composite reported in this work exhibits relatively lower photocatalytic degradation efficiency compared to composite prepared with TiO_2_ or ZnO as the inorganic phases. Therefore, in future work, we will explore the incorporation of small amounts of Ti or Zn compounds to enhance the photocatalytic performance while maintaining its magnetic properties, ultimately leading to the development of more cost-effective novel photocatalytic functional materials.

Building on our group’s previous work, the synthesis of the CQDs/γ-Fe_2_O_3_ composite further demonstrates the potential of utilizing the unique properties of various organic and inorganic waste materials to produce low-cost, high-value functional materials. Given the high costs and profit margins associated with commercial photocatalytic nanomaterials, the CQDs/γ-Fe_2_O_3_ composite holds significant promise for large-scale commercialization. Additionally, it offers a promising model for the co-valorization of diverse waste streams, aligning with the principles of sustainable development and green chemistry. This approach not only addresses the challenges associated with the low-value utilization of restaurant waste and RM but also provides an innovative pathway for the green, high-value, and sustainable resource utilization of these waste streams.

## Figures and Tables

**Figure 1 nanomaterials-15-00051-f001:**
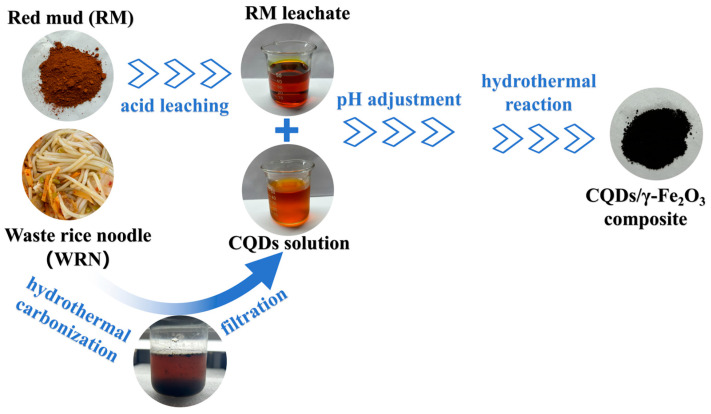
Formation process of CQDs/γ-Fe_2_O_3_ composite using WRN and RM as raw materials.

**Figure 2 nanomaterials-15-00051-f002:**
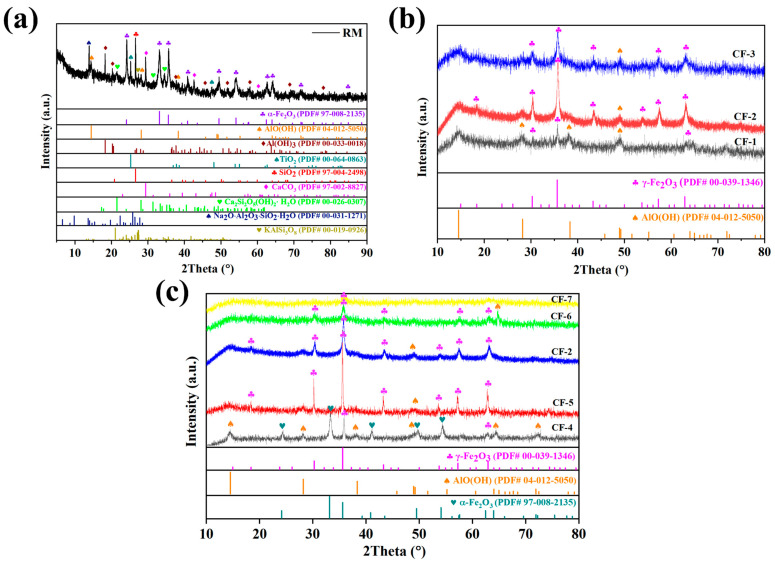
PXRD patterns of RM (**a**), CQDs/γ-Fe_2_O_3_ composites prepared under different pH conditions (**b**), and CQDs/γ-Fe_2_O_3_ composites prepared with varying CQDs dosages (**c**), with the CF-4 sample serving as a control without CQDs loading.

**Figure 3 nanomaterials-15-00051-f003:**
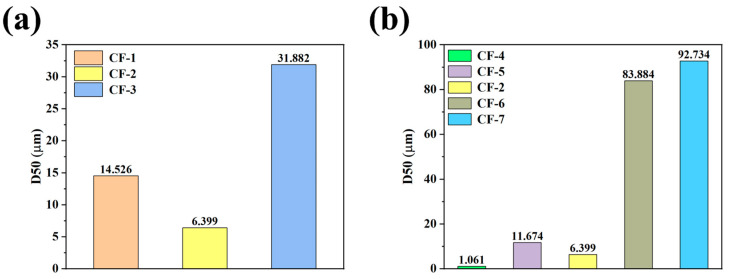
Median particle size (D50) of CQDs/γ-Fe_2_O_3_ composites prepared under different pH conditions (**a**) and with varying CQD dosages (**b**), with the CF-4 sample as a control (without CQD loading).

**Figure 4 nanomaterials-15-00051-f004:**
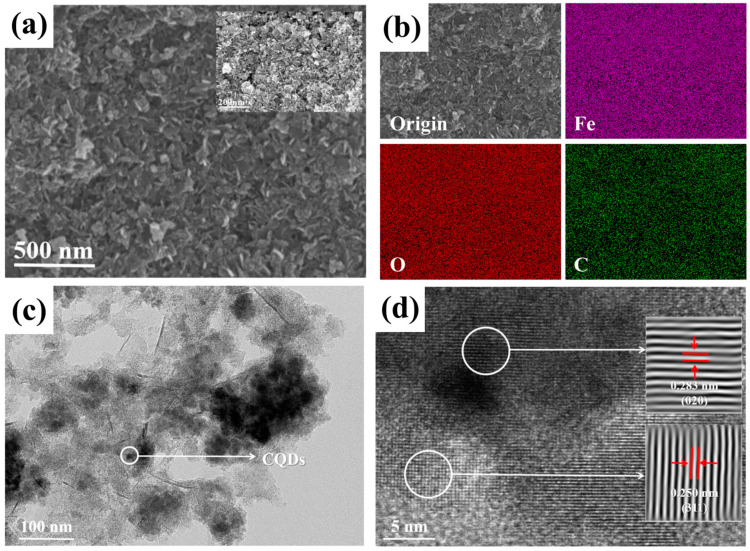
SEM image (**a**), EDS energy spectra (**b**), TEM image (**c**), and HRTEM image (**d**) of CQDs/γ-Fe_2_O_3_ composite derived from WRN and RM (CF-2 sample).

**Figure 5 nanomaterials-15-00051-f005:**
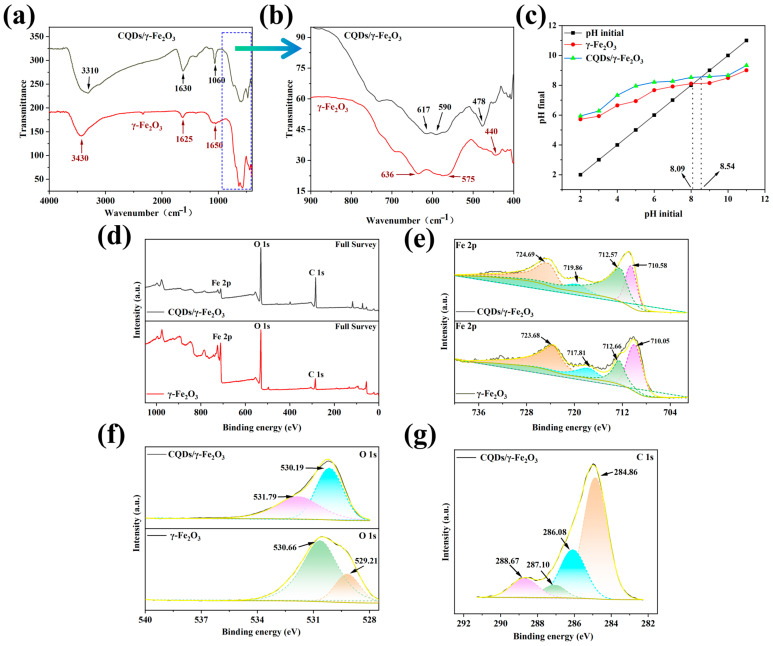
(**a**,**b**) Full spectrum (**a**) and magnified view (**b**) of the 400–900 cm^−1^ region in the IR spectra of commercial γ-Fe_2_O_3_ and CQDs/γ-Fe_2_O_3_ composite. (**c**) PZC value of γ-Fe_2_O_3_ and CQDs/γ-Fe_2_O_3_ composite; (**d**–**f**) The full XPS (**d**), high-resolution Fe 2p (**e**) and O 1s (**f**) XPS spectra of γ-Fe_2_O_3_ and CQDs/γ-Fe_2_O_3_ composite; (**g**) The high-resolution C 1s XPS spectra of CQDs/γ-Fe_2_O_3_ composite. All the CQDs/γ-Fe_2_O_3_ composites used herein were CF-2 samples.

**Figure 6 nanomaterials-15-00051-f006:**
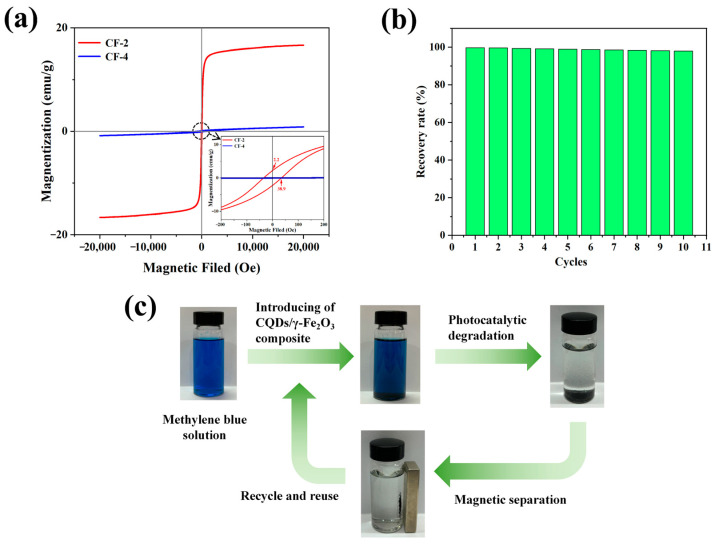
(**a**) Comparison of magnetization curves between CF-2 sample and CF-4 sample; (**b**) Recovery efficiency of CQDs/γ-Fe_2_O_3_ composite over multiple photocatalytic cycles; (**c**) Diagram depicting the magnetic separation and reuse process of CQDs/γ-Fe_2_O_3_ composite.

**Figure 7 nanomaterials-15-00051-f007:**
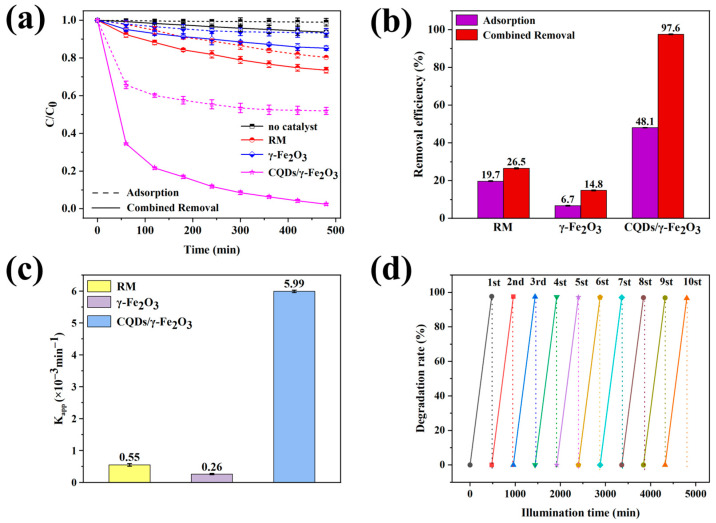
(**a**–**c**) Photocatalytic degradation curve (**a**), adsorption and combined removal efficiencies (**b**), and apparent degradation rate constants (K_app_, (**c**)) for the CQDs/γ-Fe_2_O_3_ composite (CF-2 sample), commercial γ-Fe_2_O_3_, and raw RM. (**d**) Recovery efficiency of the CQDs/γ-Fe_2_O_3_ composite (CF-2 sample) over multiple photocatalytic cycles.

**Figure 8 nanomaterials-15-00051-f008:**
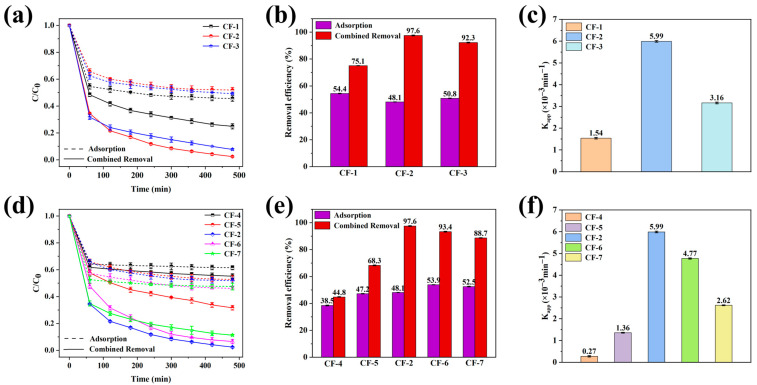
(**a**–**c**) Photocatalytic degradation curve (**a**), adsorption and combined removal efficiencies (**b**), and K_app_ (**c**) for CQDs/γ-Fe_2_O_3_ composites synthesized under varying pH levels. (**d**–**f**) Photocatalytic degradation curve (**d**), adsorption and combined removal efficiencies (**e**), and K_app_, (**f**) for CQDs/γ-Fe_2_O_3_ composites synthesized with differing amounts of CQDs.

**Figure 9 nanomaterials-15-00051-f009:**
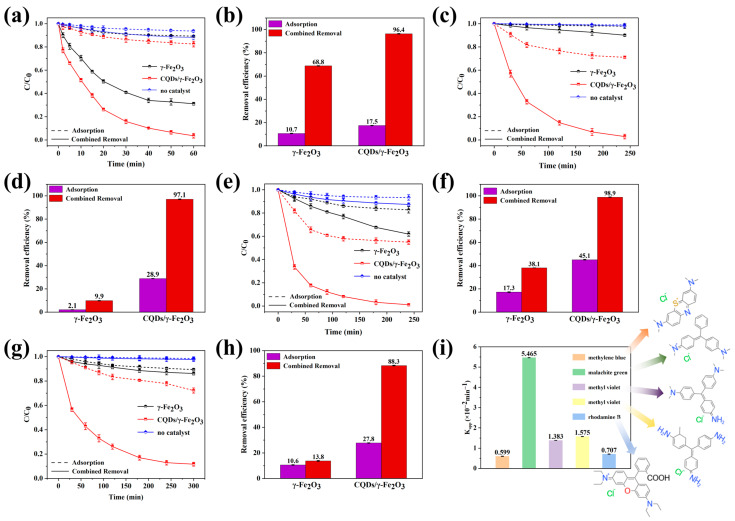
(**a**–**h**) Photocatalytic degradation performance of the CQDs/γ-Fe_2_O_3_ composite (CF-2 sample) and commercial γ-Fe_2_O_3_ for malachite green (**a**,**b**), methyl violet (**c**,**d**), basic fuchsin (**e**,**f**), and rhodamine B (**g**,**h**) under 405 nm purple light at varying irradiation times. (**i**) Comparative analysis of apparent degradation rate constants (K_app_) for five organic dyes: methylene blue, malachite green, methyl violet, basic fuchsin, and rhodamine B.

**Figure 10 nanomaterials-15-00051-f010:**
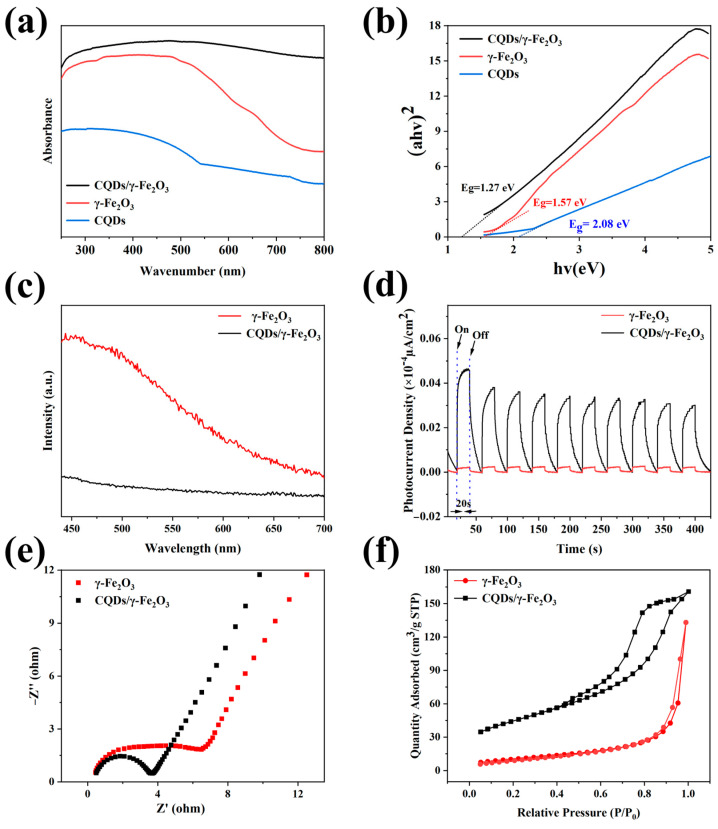
The UV-vis absorption spectra (**a**), Tauc plot curves (**b**), fluorescence spectra (**c**), photocurrent response (**d**), EIS (**e**), and BET (**f**) of the CQDs/γ-Fe_2_O_3_ composite (CF-2 sample) and commercial γ-Fe_2_O_3_.

**Figure 11 nanomaterials-15-00051-f011:**
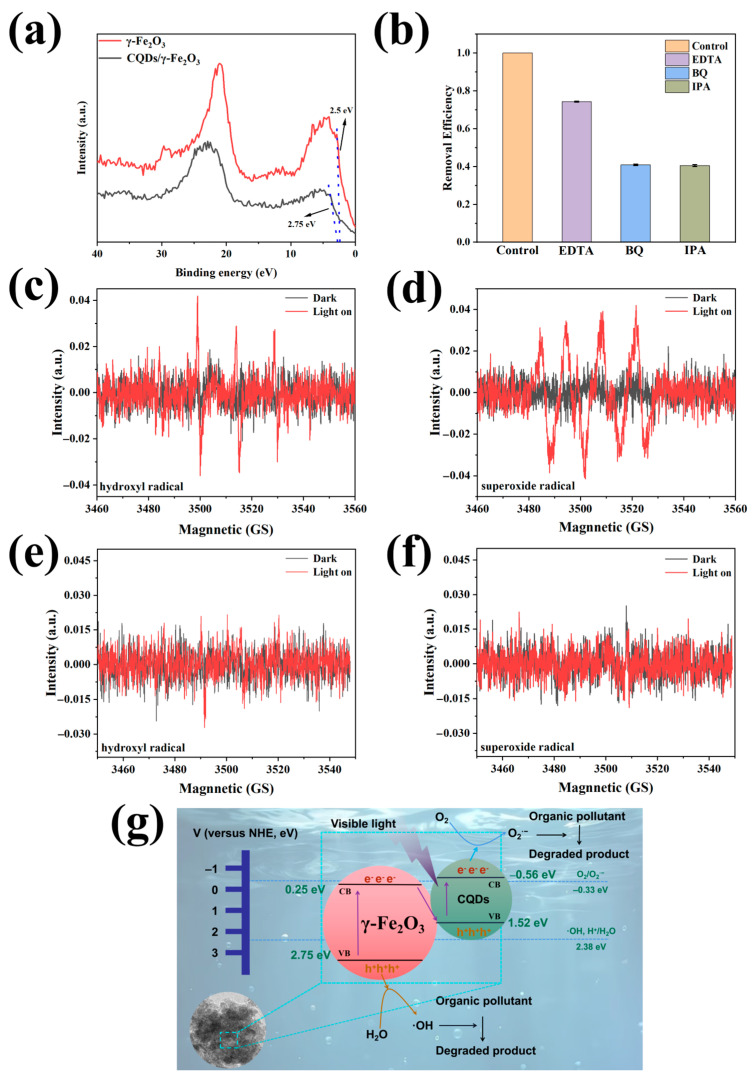
(**a**) The valence band spectra of CQDs/γ-Fe_2_O_3_ composite and commercial γ-Fe_2_O_3_. (**b**) Effects of different scavengers (EDTA-2Na, BQ, or IPA) on the photodecomposition of methylene blue under 405 nm purple light irradiation. (**c**,**d**) ESR spectra of radicals trapped by DMPO in aqueous (**c**) and methanolic (**d**) solutions of the CQDs/γ-Fe_2_O_3_ composite. (**e**,**f**) ESR spectra of radicals trapped by DMPO in aqueous (**e**) and methanolic (**f**) solutions of γ-Fe_2_O_3_. (**g**) Schematic representation of the photocatalytic mechanism for the CQDs/γ-Fe_2_O_3_ composite.

**Figure 12 nanomaterials-15-00051-f012:**
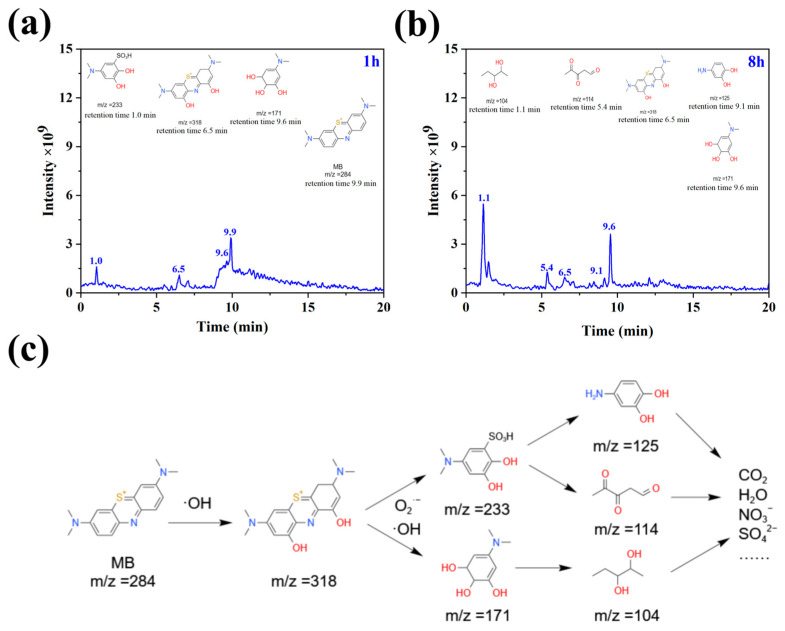
(**a**,**b**) LC-MS chromatograms of methylene blue after photodegradation using CQDs/γ-Fe_2_O_3_ composite as the photocatalyst under 405 nm purple light irradiation for 1 h (**a**) and 8 h (**b**). (**c**) Possible photocatalytic degradation pathways of methylene blue in the presence of CQDs/γ-Fe_2_O_3_ composite under 405 nm purple light irradiation.

**Table 1 nanomaterials-15-00051-t001:** Formulation design of CQDs/γ-Fe_2_O_3_ composites. The CF-4 sample served as a control without CQDs loading.

Sample Number	pH	The Volume Ratio of RM Leachate to CQDs Solution	Elemental Composition (wt%)					
			Fe	O	C	Al	Ca	Ti
CF-1	4	1:2	36.29	29.72	11.12	21.75	0.29	0.56
CF-2	8	1:2	56.45	28.68	10.03	2.72	0.65	1.41
CF-3	12	1:2	55.72	28.45	9.19	3.44	1.03	2.14
CF-4	8	1:0	47.15	32.56	-	17.74	0.82	1.66
CF-5	8	1:1	59.27	30.68	4.86	2.95	0.71	1.53
CF-6	8	1:3	52.13	26.89	16.65	2.61	0.57	1.13
CF-7	8	1:4	46.15	24.38	24.96	2.57	0.65	1.23

**Table 2 nanomaterials-15-00051-t002:** XPS peak distribution of commercial γ-Fe_2_O_3_ powder and CQDs/γ-Fe_2_O_3_ composite (CF-2 sample).

Photocatalyst	Element	Peak (eV)	Surface Group	Assignment
CQDs/γ-Fe_2_O_3_ composite	C 1s	284.86	C	Graphitic carbon
		286.08	C–O	Alcoholic or etheric structure in CQDs
		287.10	C–O–Fe	The C–O structure connecting CQDs and the surface of γ-Fe_2_O_3_
		288.67	C=C	Aromatic ring of CQDs
	O 1s	530.19	Fe–O	Oxygen bonded to iron
		531.79	C–O	Oxygen bonded to CQDs
	Fe 2p	710.58	Fe	Fe^2+^ in Fe (2p_3/2_)
		712.57	Fe	Fe^3+^ in Fe (2p_3/2_)
		719.86	Fe	Satellite peak of Fe^3+^
		724.69	Fe	Fe (2p_1/2_)
Commercial γ-Fe_2_O_3_	O 1s	530.66	Fe–O	Oxygen bonded to iron
		529.21	Fe–OH	Surface hydroxyl group of FeO_x_
	Fe 2p	710.05	Fe	Fe^2+^ in Fe (2p_3/2_)
		712.66	Fe	Fe^3+^ in Fe (2p_3/2_)
		717.81	Fe	Satellite peak of Fe^3+^
		723.68	Fe	Fe (2p_1/2_)

**Table 3 nanomaterials-15-00051-t003:** Photocatalytic degradation efficiencies of CQDs/iron oxide composites derived from different iron and carbon precursors.

Iron Source	Carbon Source of CQDs	Light Source	Pollutant	Pollutant Concentration	Photocatalyst Dosage (g/L)	Irradiation Times (min)	Degradation Rate (%)	Reference
Waste RM	WRN	Purple light lamp (20 W, wavelength: 405 nm)	Methylene blue	20 mg/L	2	480	97.6	This work
			Malachite green	20 mg/L	2	60	96.4	
			Methyl violet	20 mg/L	2	240	97.1	
			Basic fuchsin	20 mg/L	2	240	98.9	
			Rhodamine B	20 mg/L	2	300	88.3	
Waste IOS	WRN	Purple light lamp (20 W, wavelength: 405 nm)	Methylene blue	20 mg/L	2	480	99.3	[42]
Commercial γ-Fe_2_O_3_	Glucose	Xe lamp (300 W, wavelength: 455 nm)	Sulfamethoxazole (SMX)	10 mg/L	0.2	120	95	[30]
FeSO_4_·7H_2_O	Corn stover	Xenon lamp (300 W, with a wavelength range of 420–760 nm)	Tetracycline	30 mg/L	1	150	96	[26]
Commercially-available magnetic Fe_3_O_4_ nanoparticles	Glucose	Xe lamp (400 W, wavelength > 420 nm)	Methylene blue (in NaOH solution)	1 × 10^–3^ mol/L	1	30	83	[27]
FeCl_3_·6H_2_O	Citric acid and Ethylenediamine	Xe lamp (300 W, wavelength: 420 nm)	Methylene blue	20 mg/L	0.5	120	99.3	[28]
Fe(NO_3_)_3_·9H_2_O	Glucose	Simulated sunlight irradiation	Methylene blue	10 mg/L	1	240	91.5	[29]
FeSO_4_·7H_2_O	Citric acid	Xe lamp with a 420 nm cutoff filter (350 W)	Tetracycline, (0.50 mM of H_2_O_2_ was added)	20 mg/L	0.25	60	93	[25]
FeSO_4_·7H_2_O	Citric acid	Xe lamp (300 W, wavelength > 420 nm)	Metronidazole	30 mg/L	0.2	45	99.36	[67]
Fe(NO_3_)_3_·9H_2_O	Citric acid	HPMVL visible light lamp (250 W)	Oxytetracycline	10 mg/L	0.2	100	98	[68]

**Table 4 nanomaterials-15-00051-t004:** Photocatalytic degradation performance of different CQDs/metal oxide composites based on WRN.

Metal Source	Carbon Source of CQDs	Possibility of Recovery Through a Magnetic Field	Pollutant	Pollutant Concentration	Photocatalyst Dosage (g/L)	Irradiation Times (min)	Degradation Rate (%)	Reference
RM	WRN	Yes	methylene blue	20 mg/L	2	480	97.60	This work
Waste IOS	WRN	Yes	methylene blue	20 mg/L	2	480	99.30	[42]
Commercial TiO_2_	WRN	No	methylene blue	20 mg/L	4	80	99.87	[40]
Commercial ZnO	WRN	No	methylene blue	20 mg/L	2	10	98.88	[41]

## Data Availability

Data is contained within the article or Appendix A.

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
