# Peer review of "A Magnetic Photocatalytic Composite Derived from Waste Rice Noodle and Red Mud"

_nanomaterials, 2024, doi:10.3390/nano15010051_

Round 1

Reviewer 1 Report

Comments and Suggestions for Authors

The manuscript presents interesting research work. In general, the manuscript is properly organized. However, some of the descriptions need to be explained in more detail.

1.      The words that appear in the title of the manuscript should not be repeated in the keywords.

2.      As an efficient, safe, and environmentally friendly water purification technology, photocatalytic degradation utilizes specific semiconductor materials (such as TiO2[1], ZnO[2], ZnS[3], CdS[4], etc.) as catalysts…. A group of materials based on bismuth should be included 10.1007/s12034-021-02504-4

3.      What is new in this study compared to others of this type and your previously published studies regarding Red Mud? The novelty in the manuscript should be clearly highlighted. Also, the authors should include the novelty of the work in terms of cost, reliability, and performance. The main advantages and disadvantages of the present study must be added.

4.      In the introduction, remove this part of the text (See 142 Figure 1).

5.      Describe the innovation of your work and add it as a paragraph to the introduction.

6.      How do you explain the poor photocatalytic degradation performance of methylene blue for commercial γ-Fe₂O₃ compared to your synthesized material containing γ-Fe₂O₃ (CQDs/γ-Fe₂O₃ composite).

7.      Doesn't UV radiation have any effect on the cost of water treatment, so you persistently emphasize the cheap purification process (large-scale water purification projects).

8.      What is the experimental error over the photocatalytic measurement? Experimental data should be accompanied by standard deviations.

9.      In my opinion, this manuscript should be published after minor revision and additional review. Overall, the paper is good, Hopefully, the authors will follow my instructions then it may be accepted in the prestigious journal.

Reviewer 2 Report

Comments and Suggestions for Authors

The manuscript “A Magnetic Photocatalytic Composite Derived from Waste Rice Noodle and Red Mud” describes fabrication, characterization and  application of novel photocatalysts in degradation of model dye- methylene blue (MB). The objective of the work is actual  and important (with agreement with UN agenda 2030 and circular economy principles).

Specific comments:

2.2. Synthesis: it is written: “The primary ionic components were as follows: Fe 26.53 g·L¹, Ti 4.66 g·L¹, Al 9.67 g·L¹, Ca 2.56 167 g·L¹, and Na 3.57 g·L¹. The resulting RM leachate and CQDs solution were mixed in a specific volume ratio and stirred for 0.5 hours, after which the pH was adjusted using ammonia or NaOH solution.”

Questions:

1.      What was the method used for quantification of Fe, Ti, Al, Ca and Na?

2.      pH was adjusted to what value?

Table 1:

“Serial number” I think that it will be better to call: sample number, or sample or name.  

Lines 197-199 “The diffraction peaks at 30.2°, 35.6°, 37.2°, 43.3°, 57.3°, and 62.9° corresponded to the (220), (311), (222), (400), (511), and (440) crystal planes of γ-FeO (maghemite, PDF card No. 00-039-1346), indicating its potential as a practical magnetic photocatalyst” please clarify

Lines 201-203 “Moreover, we observed that pH selection was a critical factor for the successful conversion of the red mud leachate to γ-FeO. The initial RM leachate was highly acidic, and without adjusting the pH, no solid product could be obtained by directly reacting it with the CQDs solution.” that´s obvious, don´t you think?

What does mean: “purity was poor”, “relatively pure”, “weakly magnetic” please be more specific.

Line 214: “On the other hand, the presence of CQDs was crucial for ensuring the formation of the γ-FeO phase”, could you please explain it?

Fig.2 (a-c). the values of 2 theta are missing

Lines 239-240: “The infrared spectrum showed that the surface of the CQDs/γ-FeO composite contained a large number of hydroxyl groups originating from CQDs.” Why the authors exclude the possibility of water adsorption?

Lines 242-244: “Compared to pure 242 γ-FeO, the FeO stretching vibration peak in the CQDs/γ-FeO composite exhibited a 243 significant redshift to 478 cm¹ and 590 cm¹,….” The compared γ-FeO, was a commercial one or it was prepared by the authors. I can see nothing in the Fig 4a. Please insert zoom of the M-O zone.

Line 248: “….due to the presence of organic groups such as hydroxyl… ” please explain

The XPS spectra are badly deconvoluted. The comparison of Fe2p spectra + deconvolution for neat Fe2O3 and CQD/Fe2O3 composite is missing.

Lines 278-281: “RM exhibited extremely weak magnetism, lacking any practical capability for magnetic field recovery. In contrast, due to the introduction of CQDs enabling the formation of the magnetic γ-FeO phase, the CQDs/γ-FeO composite material exhibited excellent soft magnetic properties.” I think I do not understand, there is nearly any gamma-Fe2O3 in RM “raw”. The γ-FeO is formed hydrothermally in the presence, or not, of CQD (sample CF-4). Why the authors expected magnetism of raw RM? In my opinion the magnetic properties of the sample CF-4 should be compared with others CQDs/γ-FeO to support the authors statement. Please clarify

Generally, the characterization of the novel materials is not completed, in my opinion. The Raman characterization is needs as for the carbon CQD as for the Fe2O3 and CQD/Fe2O3.

Photocatalytic activity

In my opinion RM has no photocatalytic activity, the reduction on the MB concentration in the presence of RM should be related to its adsorption properties. The dark experiment during 480 min is missing. Moreover all the materials should be tested in the dark during 480 min., but not for less than 100 min. The curve profiles presented in Fig 6 b,e, and g in my opinion support this statement that the materials adsorbed the MB but not photodegraded. This has to be clearly proved – the influence of light during the experiment (the same duration time as for adsorption process – in the dark as for photocatalytic performance -under light irradiation should be presented as 2 independent experiments). The 8 hours is a vary long time for photocatalytic dye degradation.

Here again the authors were using raw RM as a reference in my opinion RM is not a “reference” for this studies, the RM was used as a source of Fe3+ only, am I right?. Or I do not understand the idea, could you please explain?

Do the authors related the Fe2O3 crystallites size in CQD/Fe2O3 materials, and photocatalytic performance, what about the Fe2O3 loading influence? What about the localization of Fe2O3 in the materials vs. catalytic activities? Could the authors comment it?

Do the authors characterize the reaction medium after photocatalytic cycle by for example TOC, NMR or other methods?

Which composite was studied in photocatalytic degradation of other dyes (RhB, MV, MG and others)?

For the band gap study the commercial  gamma-Fe2O4 was used or the sample CF-4?. The studies were performed for which composite?

I do not understand how the authors calculated VB and CB potential from the XPS. Some literature references may be?

Based on the above comments, I cannot support this manuscript for publication until the necessary corrections have been made

Round 2

Reviewer 2 Report

Comments and Suggestions for Authors

The manuscript “A Magnetic Photocatalytic Composite Derived from Waste Rice Noodle and Red Mud” describes fabrication, characterization and application of novel photocatalysts in degradation of model dye- methylene blue (MB). The objective of the work is actual (with agreement with UN agenda 2030 and circular economy principles) and important.

Thank you for the detailed answers to my questions; however, there are still some points that need clarification.

Specific comments:

The XPS spectra are still badly deconvoluted. The comparison of Fe2p spectra + deconvolution for neat Fe2O3 and CQD/Fe2O3 composite is missing.

1.      The experimental curve in the O1s region and in C 1s region does not align with the theoretical curve.

2.      Deconvolution of the XPS spectrum in Fe 2p region: the authors used very asymmetric band for Fe2+ “component” why? Why were satellite bands not considered in the deconvolution process?

 The FTIR spectra are better quality now, thank you. But still in Fig 5 a, black curve, I see 2 bands at ca. 590 cm-1. There as the authors mention one band only. Those bands show “better” split for g-Fe2O3 (red curve) the two bands are noticeable at 575 cm-1 and other in my opinion at 590 cm-1, thus I have a doubt if the authors compared proper and band position.

 Thank you for including Raman spectra, but I agree that they are not informative, as you suggested, due to storage of the sample.

Based on the above comments, I cannot support this manuscript for publication until the necessary corrections have been made
